# Commercial Methods for Antifungal Susceptibility Testing of Saprophytic Molds: Can They Be Used to Detect Resistance?

**DOI:** 10.3390/jof10030214

**Published:** 2024-03-14

**Authors:** Paschalis Paranos, Ana Espinel-Ingroff, Joseph Meletiadis

**Affiliations:** 1Clinical Microbiology Laboratory, Attikon University Hospital, Medical School, National and Kapodistrian University of Athens, 12462 Athens, Greece; pasxalisparanos@hotmail.gr; 2Virginia Commonwealth University Medical Center, Richmond, VA 23219, USA; victoria.ingroff@vcuhealth.org

**Keywords:** Antifungal Susceptibility Testing, *Aspergillus*, Sensititre YeastOne, Etest, Micronaut, VIP check

## Abstract

Commercial tests are often employed in clinical microbiology laboratories for antifungal susceptibility testing of filamentous fungi. Method-dependent epidemiological cutoff values (ECVs) have been defined in order to detect non-wild-type (NWT) isolates harboring resistance mechanisms. We reviewed the literature in order to find studies where commercial methods were used to evaluate for in vitro susceptibility of filamentous fungi and assess their ability to detect NWT isolates according to the available ECVs. Data were found for the gradient concentration strips Etest and MIC Test Strips (MTS), broth microdilution Sensititre YeastOne (SYO), Micronaut-AM and the agar dilution VIPcheck assays. Applying itraconazole, voriconazole and posaconazole Etest ECVs for *A. fumigatus*, Etest was able to detect 90.3% (84/93), 61.2% (90/147) and 86% (31/36) of isolates with known *cyp51A* mutations, respectively. Moreover, Etest also was able to detect 3/3 *fks* mutants using caspofungin ECVs and 2/3 micafungin mutant isolates. Applying the voriconazole and posaconazole SYO ECVs, 57.7% (67/116) and 100% (47/47) of mutants with known *cyp51A* substitutions were classified as NWT, respectively. VIPcheck detected 90.3% (159/176), 80.1% (141/176) and 66% (141/176)of mutants via itraconazole, voriconazole and posaconazole, respectively, whereas Micronaut-AM detected 88% (22/25). In conclusion, Etest posaconazole and itraconazole, as well as micafungin and caspofungin ECVs, detected *A. fumigatus* mutants. On the other hand, while the posaconazole SYO ECV was able to detect *cyp51A* mutants, similar data were not observed with the SYO voriconazole ECV.

## 1. Introduction

The prevalence of invasive fungal infections continues to increase due to immunocompromised individuals. In order to manage these infections three classes of antifungal agents (echinocandins, azoles and polyenes) are recommended as first-line or salvage therapy [1]. Given the emergence of isolates with intrinsic or acquired resistance associated with high mortality, the Clinical and Laboratory Standards Institute [CLSI] and the European Committee on Antifungal susceptibility Testing [EUCAST]) have developed standardized broth microdilution methods for in vitro antifungal susceptibility testing. The EUCAST method utilizes 96-microtiter flat bottom plates, RPMI medium containing 2% glucose buffered with MOPS, a 10^5^ CFU/mL inoculum, visual and/or spectrophotometric determination of minimal inhibitory concentrations (MIC) for azoles and amphotericin B as the lowest drug concentration with >90% growth inhibition and minimal effect concentrations (MEC) for echinocandins as the lowest drug concentration with abnormal, short, and branched hyphal clusters, whereas the CLSI method utilizes 96-microtiter U-shaped plates, RPMI medium containing 0.2% glucose buffered with MOPS, a 10^4^ CFU/mL inoculum and visual determination of MIC for azoles and amphotericin B corresponding to the lowest drug concentration with complete growth inhibition and MEC for echinocandins as the lowest drug concentration with small, rounded compact hyphal forms [2,3]. Species-specific breakpoints (BP) have also been established for interpreting MIC/MECs of some antifungal agents against the most prevalent species [4,5,6]. Results by reference methods have been correlated with in vivo outcome, as infections by azole resistant *Aspergillus* isolates have been associated with increased mortality [7]. Amphotericin B failure has been linked with isolates with high MICs [8,9] and micafungin therapy failed against an *Aspergillus* isolate with reduced susceptibility to echinocandins [10], while several preclinical models show the importance of reference MICs regarding in vivo outcome [11].

### 1.1. Development of Clinical Breakpoints and ECVs for Filamentous fungi

The role of antifungal susceptibility testing relies on the ability to select the most appropriate agent for the treatment of a specific fungal infection. Even though methodological differences exist between CLSI and EUCAST procedures, their results have proven to be comparable [12] and allow the categorization of the strains as susceptible or resistant by applying the established BPs. EUCAST has defined drug- and species-specific clinical BPs for *Aspergillus* spp. versus triazoles (itraconazole, posaconazole, voriconazole and isavuconazole) and amphotericin B [13], whereas the CLSI has recently adopted a clinical BP only for voriconazole and *A. fumigatus* [14]. Although BPs can predict the likelihood of clinical response to antifungal therapy, there are many species and antifungal drugs for which there are insufficient data to determine clinical BPs [4]. For those species and drugs, epidemiological cutoff values (ECOFFs for the EUCAST reference method and ECVs for the CLSI reference method) can be used in order to detect isolates with acquired resistance mechanisms [5,15]. 

An ECV is defined as the highest MIC of the wild-type (WT) population of a given species without a phenotypically detectable acquired resistance mechanism [4]. The main role of an ECV is to distinguish WT isolates from non-wild-type (NWT) isolates [16,17,18], i.e., isolates with MIC higher than the ECV that potentially harbors a known or unknown acquired resistance mechanism [19]. Moreover, ECVs have an important role in tracking MIC elevation and emergence of resistance. ECVs are determined based on MIC distributions integrating information from with drug resistance mechanisms whenever available, whereas BPs are based on data for ECVs, pharmacokinetic/pharmacodynamic studies and correlation of the MIC with clinical outcome [20,21,22]. Therefore, the NWT or WT is not equivalent with the terms “susceptible” or “resistant” [14] to an antifungal agent. For some species, WT isolates may naturally possess resistance mechanisms, in which case the species is considered intrinsically resistant; for example, *Fusarium* spp. and several drugs [14]. Thus, WT isolates may or may not respond to antifungal therapy, whereas NWT isolates are expected to be associated with clinical failure [23]. The development of ECVs is dependent on the in vitro susceptibility testing used to generate MIC values. ECVs have been published for the most common *Aspergillus* species [5,15], and for some *Fusarium* and *Zygomycetes* spp. [14,24] mainly for the reference methodologies (Table 1).

CLSI and EUCAST followed a strict ECV setting process in order to determine “reference” ECVs based on MIC data generated with reference methods [32]. Apart from “reference” ECVs, there are also “method-dependent” ECVs for commercial susceptibility testing methods, and particularly Etest for *Aspergillus* spp. [25,29]. Unfortunately, apart from the most common *Aspergillus* spp., “method-dependent” ECVs have not been defined for other filamentous fungi (Table 1). Considering the comparative rarity of infections caused by less prevalent molds, it may take years before sufficient reliable data will be available to establish ECVs for available commercial methods.

### 1.2. Commercial Methods for Antifungal Susceptibility Testing of Molds

Reference antifungal susceptibility testing methods are not widely used because they are difficult to implement and require expertise. For optimal patient management and for routine practice laboratories, antifungal susceptibility methods should be fast, accurate, user-friendly, reproducible and low-cost. Across the years, several commercially available antifungal susceptibility methods have been used in clinical and research laboratories [33]. These methods could be helpful in such limited settings for MIC/MEC determination. Commercially available susceptibility methods have been compared with reference assays in two ways: the essential agreement (EA), which is the agreement between MICs/MECs of reference and commercial methods (usually within 1–2 twofold dilutions), or the categorical agreement (CA), which is the agreement between categorization of isolates as susceptible, intermediate or resistant with commercial and reference methods [19,26]. Therefore, commercially available and ready-to-use methods could be a better alternative for the routine clinical microbiology laboratory as far as they are able to produce similar results with reference standards [26]. 

In particular, five commercial tests are available for screening antifungal resistance of moulds: two broth microdilution-based methods, the Sensititre YeastOne (SYO, TREK Diagnostics System, Cleveland, OH, USA) and the Micronaut AM (Bruker, Billerica, MA, USA) and three agar-based methods using strips with a gradient of antifungal concentrations, the Etest (Biomerieux, Tokyo, Japan), MIC Test Strips (MTS) (Liofilchem, Roseto degli Abruzzi, Italy) and the four-well plates VIPcheck^TM^ (Mediaproducts BV, Groningen, The Netherlands) (Figure 1) [34,35]. Although they are easier compared to reference methods, they are expensive, and MIC reading can be difficult because of subtle color changes, trailing growth and isolated colonies, and may not perform equally well for all species and drugs. Most of these tests have been developed for yeasts and then applied to molds. For these reasons, commercial tests needs to be calibrated based on reference methods by using QC of reference methods, generating similar MIC distributions for WT isolates and detecting NWT isolates with different levels of resistance. The agreement between commercial and reference methodologies, together with the two-fold differences between median MICs, is summarized in Table 2. A more-than-one two-fold difference between commercial and reference methods indicates a significant difference in MIC distributions for the two methods that could lead to classification errors if reference ECV/BPs will be used.

Concerning colorimetric methods, SYO yielded high essential agreement with the CLSI for voriconazole, posaconazole and itraconazole, except for *A. fumigatus* and *A. nidulans* isolates [37], while for the echinocandins, high essential agreement was found only for micafungin versus *A. fumigatus* and *A. flavus* isolates [36] (Table 2). Results for amphotericin B also showed high essential agreement (91.3–100%) depending on *Aspergillus* spp., but not for the zygomycetes. Reliable alternatives for antifungal susceptibility testing for *Aspergillus* and non-*Aspergillus* species are Etest and MTS: categorical and essential agreements of ≥90% with CLSI and EUCAST [40,45]. Briefly, concerning echinocandins for both micafungin and caspofungin, good essential agreement (>77.8%) was found for all species except *A. glaucus*, *Scedosporium apiospermum* and *Scopulariopsis* spp. High essential agreement was reported for the azoles and the polyenes, in particular for amphotericin B, posaconazole and voriconazole, but not itraconazole (>75%), although in some studies lower essential agreement was found. Results for Micronaut-AM in one study demonstrated good essential agreement with the CLSI reference method (>90%) for anidulafungin, voriconazole and amphotericin B, but not for itraconazole (87%) for 78 *Aspergillus* isolates [48]. In another study, low essential agreement (<62%) was found for all triazoles, but not for amphotericin B (100%) for 77 *Aspergillus* isolates with the EUCAST reference method [49]. Finally, categorical agreement was also stated for VIPcheck^TM^ as the method can discriminate NWT isolates of *A. fumigatus*. Overall, good categorical agreement was found in two studies for all triazoles (80–97.8%), except in one study for posaconazole where agreement was 73.6% [50,51]. 

Difficulties in susceptibility testing of molds and the issue of lack of clinical data preclude the definition of BPs for commercially available methods, but method-specific ECVs have been defined for various *Aspergillus* spp. and antifungal drugs [4,16,29]. However, ECVs of commercial methods may differ from ECVs of reference methods when MIC distributions obtained using each method are different (Table 1). Although for most drug-species the differences are within one two-fold dilution, there are notable exceptions like Etest with caspofungin and posaconazole against *A. terreus*, voriconazole against *A. flavus*, posaconazole against *A. niger*, as well as amphotericin B against *S. apiospermum* and *Scopulariopsis* spp. and for SYO with caspofungin against all *Aspergillus* spp. and for amphotericin B against the zygomycetes. Similarly, Micronaut-AM MICs of all three triazoles were 2–3 two-fold dilutions different from EUCAST but not from CLSI MICs (except itraconazole) for *Aspergillus* spp. [40,45]. These results lead to susceptibility classification errors when established BPs or ECVs for reference methods are used for interpreting MICs generated with commercial methods.

## 2. Purpose of Review

The main purpose of the current review was to summarize the ability of commercial methods for in vitro susceptibility testing of filamentous fungi to detect NWT isolates to the triazoles, echinocandins or polyenes using available ECVs. Based on the role of the ECV to detect mutants [52], we focused on publications where data on resistance mechanisms are presented together with the MIC of the isolates for particular species and drugs [53]. For this reason, we initially describe the mechanism of actions and resistance and summarize known mutations in target and other genes that are associated with resistance. As expected, data for non-*Aspergillus* species were scarce and most publications reported only Etest and SYO data. The importance of validating these methods as predictors of in vitro resistance is deemed necessary as, apart from reference laboratories, the majority of clinical microbiology routine laboratories use commercial methods for in vitro susceptibility testing of molds [33]. However, as is the case for any susceptibility test, ECVs can provide incorrect classification or overlapping results between mutants and WT isolates as reported elsewhere [33]. For instance, MICs of mutants may be lower than the given ECV for particular agents and species. Although this might be true for mutations that do not elevate MICs, for mutations that are known to confer resistance, any misclassification would mean failure of the test and further optimization may be needed.

### 2.1. Echinocandins

Echinocandins target the fungal cell wall via non-competitive inhibition of (1,3)-β-D glucan (BDG) and—contrary to *Candida* spp., which is fungicidal—present fungistatic activity against *Aspergillus* spp. and other filamentous fungi. Despite the understanding of resistance mechanisms for *Candida* isolates, they are not as well documented for *Aspergillus* spp. There is minimal data for the treatment of invasive aspergillosis because echinocandins have been used mainly as salvage or combination therapy. Differences in innate echinocandin susceptibility show up in *A. niger* due to different cell wall composition [54]. *fks*1 gene substitutions, the effect of chitin synthesis, genetic repression in heat shock protein 90 (Hsp 90) and the reactive oxygen species (ROS) are among the molecular mechanisms of echinocandin resistance reported in *Aspergillus* spp. isolates [35,55]. 

Amino acid substitutions in the *fks*1 subunit of glucan synthesis lead to echinocandin resistance. Although knowledge is scarce about echinocandin resistance among *Aspergillus* spp., an expression analysis of the *fks*1 gene revealed an overexpression (three-fold higher) in comparison with a WT isolate, after a treatment failure with caspofungin in an invasive aspergillosis infection caused by *A. fumigatus* [56]. In order to understand echinocandin resistance, multiple in vitro models with resistance induction have been developed [57,58]. Briefly, site-directed mutation at S678Y into *fks*1 led to decreased susceptibility to caspofungin with MEC = 4 mg/L [57], while substitution of serine with proline in codon 678 resulted in a resistance phenotype of MEC ≥ 16 mg/L to all the three available echinocandins. This result suggests that *fks*1 gene modifications lead to echinocandin resistance [58]. Previously, an echinocandin-resistant *A. fumigatus* isolate harboring the point mutation F675S in the *fks*1 gene had reduced susceptibility to both caspofungin and micafungin (MEC = 2 mg/L) [10]. However, an *fks*1-independent mechanism of echinocandin resistance in *A. fumigatus* has been recently identified: an alteration of the drug–target interaction via caspofungin-induced ROS-mediated changes in the lipid composition of the glucan synthase with elevated MECs (caspofungin (4–16 mg/L) and micafungin (2–4 mg/L)) [59].

Another resistance mechanism of *Aspergillus* species to echinocandins is the effect of chitin synthesis. The paradoxical phenomenon or the ability of *Aspergillus* spp. isolates to grow in concentration above MEC is related to the increased chitin synthesis in the fungal cell wall. This phenomenon is more commonly seen with caspofungin than micafungin and anidulafungin [60]. Increased sensitivity of caspofungin mutants was in agreement with synergistic antifungal effect of a combination of chitin synthesis inhibitor and caspofungin, while Δ*ras* mutants, despite having a low level of β-glucan, are more resistant to caspofungin due to an increase in the cell wall synthesis [61]. Increased sensitivity to caspofungin was also observed among calcineurin mutants (Δ*cnaA*) of *A. fumigatus* isolates, due to low level β-glucan and chitin synthesis [62]. 

Hsp 90 has also been implicated in echinocandin resistance. It is involved in a wide range of signaling networks and cell processes, from control to survival of the cell cycle, as well as response to cell stress in order to maintain cell homeostasis [63]. Hsp 90 plays a key role in the evolution of azole and echinocandin resistance by activating specific cellular signaling pathways that are necessary for cell survival against membrane stress due to the antifungal agent [64]. More specifically, resistance to echinocandins is affected through calcineurin, a protein phosphatase regulator of cellular signaling. However, genetical repression of Hsp 90 leads to decreased virulence in a murine infection model of IA; the replacement of natural promoters with two artificial promoters in *A. fumigatus* isolates resulted in increased susceptibility to caspofungin and a canceling of the paradoxical effect [65].

Finally, a new mechanism was described via ROS production. It was found that caspofungin exposure modifies glucan synthase, rendering it insensitive to echinocandins. This mechanism of resistance involved alteration of the glucan synthase lipid microenvironment and was mediated via an off-target effect on mitochondria leading to increased ROS. Thus, it was hypothesized that caspofungin-induced ROS alters the lipid composition around glucan synthase, changing its conformation and making it insensitive to echinocandins [66]. The resistance mechanisms for other filamentous fungi have not been fully explored.

### 2.2. Triazoles

Azole resistance is usually associated with specific resistance mechanisms constituted by a variable number of tandem repeat (TR) integrations in *cyp*51A promoter and mutations in the coding gene [35]. Isolates with resistance to azoles due to TR (TR_34_/L98H, TR_34_/L98H/S297T/F495I, TR_46_/Y121F/T289A and TR_53_) have been detected throughout the world [67]. Moreover, single nucleotide polymorphisms, mainly in gene positions G54, M220 and G448 of the *cyp*51A gene, have been observed to be more frequent in patients with chronic pulmonary aspergillosis, long term azole therapy, and clinical treatment failures [35]. Single-point mutations in other positions have also been associated with azole resistance (G138C, F219I, P216L, G432S and G432A) [68,69,70]. Isolates with substitutions in *cyp*51A usually have high MIC values for voriconazole (0.5–>32 mg/L), itraconazole (0.5–32 mg/L), posaconazole (0.125–>16 mg/L) and isavuconazole (0.125–>16 mg/L) [45], depending on the mutation [35]. It is widely accepted that TR_34_/L98H alterations are associated with pan-azole resistance, and TR_46_/Y121F/T289A with a high level of voriconazole resistance but with variable posaconazole and itraconazole susceptibility. M220 alterations are associated with resistance to itraconazole and posaconazole and with variable susceptibility to voriconazole, except M220T which shows susceptibility to posaconazole and voriconazole. Similarly, G54 alterations are associated with resistance to itraconazole and posaconazole but not to voriconazole [71].

Although mutations in *cyp51A* have been well explored, there are also mutations or overexpression in *cyp51B* and *cyp51C* which confer resistance to the triazoles. Most *Aspergillus* spp., *Mucorales* spp. and *Penicillium* spp., have two paralogues (*cyp51A* and *cyp51B*), while there are few species, including *A. flavus*, *A. oryzae* and the *Fusarium* spp., that have three *cyp51* enzymes (*cyp51A, cyp51B* and *cyp51C*) [72,73]. However, while mutations in *cyp51B* responsible for azole resistance in *A. fumigatus* have not been reported, overexpression in two clinical azole-resistant isolates suggest its possible role [74]. Interestingly, in a single study, a novel G457S mutation in *cyp51B* with concomitant F390L mutation in the 3-hydroxy-3-methylglutaryl-coenzyme-A-reductase-encoding gene (hmg1) showed high MICs in itraconazole, voriconazole, posaconazole and isavuconazole (>8 mg/L). The contribution of azole resistance is unclear due to both mutations [72], while reconstitution of the G457S mutation in a triazole-sensitive strain resulted in resistance to voriconazole (2 mg/l), but not to itraconazole or posaconazole [75]. Four mutations of *A. flavus* in *cyp51C* (S196F, A324P, N423D and V465M) are correlated with voriconazole resistance [76]. Moreover, it has been found that T788G missense mutation in *cyp51C* gene was responsible for a voriconazole-resistant *A. flavus* clinical isolate (MIC = 8 mg/L) [77]. Finally, in a resistant *A. flavus* isolate it has been reported that substitution in H349R in *cyp51C* showed increased gene expression but the role of azole resistance remained unclear [78].

Apart from mutations in *cyp51-*related genes, there are also resistance mechanisms that involve resistance to azoles among *Aspergillus* spp. isolates (overexpression of efflux pumps, upregulation of *cyp51A*, CCAAT-binding complex, upregulation of efflux pumps, hmg1 mutations, master regulators, damage resistance protein 1, mismatch repair gene (MSH2), OrmA enzyme, deletion of the CybE encoding gene (b5 CybE), oxidoreductase HorA, hapE, *A. fumigatus* farnesyltransferase Cox10 (Afcox10), RNA interference (RNAi)-dependent mutations, cholesterol uptake/import resistance mechanisms and cytochrome c oxidase cox7c W56* nonsense) [35,64,79]. Efflux pumps, particularly major facilitator superfamily (MFS) and ATP-binding cassette (ABC) transporters, remove toxins by coupling transport with proton gradient or adenosine triphosphate (ATP) hydrolysis [64]. In itraconazole-resistant *A. fumigatus* isolates, overexpression of ABC transporters (*AfuMDR1*, *AfuMDR2* and *AtrF*) and upregulation of *AfuMDR3* and *AfuMDR4* encoding MFS-transporters have been described [80,81]. Moreover, upregulation in transporter genes (*abcB*/Afu1g10390, *abcE*, *mfsA*, *mfsB* and *mfsC*) [82] and *cdr1B* [83] were shown in response to voriconazole and itraconazole exposure in azole-susceptible and azole-resistant *A. fumigatus* isolates, respectively. Increased expression of *cdr1B* resulted in voriconazole MIC = 1 mg/L, posaconazole MIC = 0.25 mg/L and itraconazole MIC = 2 mg/L [83]. In addition, *hapE* is an important subunit in the CCAAT-binding complex, which plays a regulatory role of fungal phenotypes in azole resistance. It was found that six non-synonymous mutations were identified in the non-coding regions, of which resistance in the progeny was due to mutation in the *hapE* gene [84]. Isolates harboring these mutations in the *hapE* gene showed increased MICs to azoles (voriconazole, itraconazole and posaconazole MIC = 2–4 mg/L, >16 mg/L and 0.25–0.5 mg/L, respectively) [84]. Furthermore, a novel mutation (R243Q) in *Afcox10* gene was shown by next-generation sequencing analysis to confer cross-resistance to itraconazole (MIC = 8 mg/L), terbinafine (MIC = 16 mg/L) and bifonazole [85]. Finally, it has been found that mutation or deletion of cox7c results explicitly in antifungal resistance to targeting enzymes, including triazoles with high MICs of voriconazole (4 mg/L), itraconazole (2 mg/L) and posaconazole (2 mg/L) in comparison with the parental strains lower MICs [79]. The resistance mechanisms for the triazoles’ other filamentous fungi have not been fully explored.

### 2.3. Polyenes

Amphotericin B binds to ergosterol in the fungal cell membrane, leading to pore formation in the cell membrane with ion leakage and consequently cell death [86]. The main driver of amphotericin B efficacy seems to be interference with the mitochondria from ROS/anti-ROS [87]. Although amphotericin B resistance mechanisms are not well understood, the oxidative injury by ROS has been also implicated in this resistance [88]. Resistance to amphotericin B (MIC ≥ 2 mg/L) has been reported for *A. fumigatus, A. niger, A. flavus, A. lentulus, A. terreus* and *A. ustus* [64,89]. In a study, resistance to polyenes was attributed to the depletion of ergosterol due to diminished binding to the cytoplasmic cell membrane and increased amphotericin B MICs (16 mg/L) [90]. According to a large review conducted last year, the pooled mean prevalence of amphotericin B resistance was 0.17% among 26,909 *Aspergillus i*solates [89]. Overall, the development of resistance to amphotericin B is rare due to its action as a rapid fungicidal agent that inhibits the fungal growth by physiochemical reaction rather than enzyme inhibition [64]. There are no studies reporting genetic mutational changes leading to increasing amphotericin B MICs. Genetic analysis of amphotericin B resistant *A. fumigatus* isolates identified missense variants in genes tcsB, mpkC and catA and mutations in fumarylacetoacetate hydrolase associated with amphotericin B resistance [91,92]. There are observations of resistant *Aspergillus* spp. isolates that failed treatment with amphotericin B, in particular an 88% mortality rate among patients treated with amphotericin B therapy for *A. flavus* infections [93]. Some reports indicate that alterations of the fungal cell wall show a correlation with amphotericin B resistance. In another study, an experimentally evolved *A. flavus* isolate was able to grow at concentrations of up to 100 μg/mL and the authors assumed that alterations in the cell wall contributed to the resistance [94]. In addition, preclinical and clinical studies showed that amphotericin B is a poor therapeutic option (96% mortality) for *A. terreus* isolates as it is intrinsically resistant [95], with most isolates exhibiting MIC values ≥ 2 mg/L [96]. To date, there is no *A. terreus* specific genomic feature that has provided an explanation for amphotericin B resistance/tolerance mechanisms [87]. In general, resistance in section *Terrei* is associated with modulating molecular chaperons, targeting ROS via mitochondria and shaping the cellular redox homeostasis [87]; while underlying mechanisms may be associated with the level of catalase production of this species, in comparison with *A. fumigatus* [97,98]. The resistance mechanisms for polyenes and other filamentous fungi need to be fully explored.

## 3. Gradient Concentration Strips

Triazoles

ECVs for Etest have been determined for itraconazole, posaconazole and voriconazole and the most common *Aspergillus* species (*A. fumigatus, A. flavus, A. terreus, A. niger, A. nidulans*) in a multicenter study (Table 3) [29]. An ECV has not been determined yet for isavuconazole and therefore the upper MIC level of WT isolates (WT-UL) was used to assess *A. fumigatus* isolates with known *cyp*51*A* substitutions. Information about resistance mechanisms of NWT isolates exists only for *A. fumigatus* and mainly involves *cyp*51*A* substitutions. Applying the Etest itraconazole ECVs for *A. fumigatus* (2 mg/L), Etest was able to detect 78/81 isolates with known *cyp*51*A* substitutions [29]. The three strains characterized as WT were two strains with G448S and I301T and one strain with M220K substitutions in *cyp*51*A* [29]. In another study, Etest was able to detect 6/6 of mutant isolates to itraconazole and 3/6 to voriconazole [99], but unable to detect any of the three isolates tested with *cyp*51*A* substitutions for all triazoles [100]. Applying isavuconazole WT-UL (2 mg/L) determine in 40 WT isolates, 72.4% (21/29) of isolates with M220 and TR mutations were categorized as NWT (21/39), while isolates harboring G54 alterations had lower MIC values and 0/10 were detected [45]. Concerning Etest and voriconazole, results obtained from different studies were somewhat less promising in the detection of mutant isolates using an Etest ECV of 0.5 mg/L. Etest was able to detect 49/75 isolates with *cyp*51*A* substitutions [29]. Among the isolates that have been characterized as WT based on the proposed ECV (25/75), there were mutants with TR_34_ (0.125–0.5 mg/L, 3/38), G54E/R/W (<0.06–0.5 mg/L, 12/12), M220I/K//R/T/V (0.125–0.5 mg/L, 8/11), G138C (0.25 mg/L, 1/1) and I301T (<0.06 mg/L, 1/1) substitutions [29]. It is worth noting that, in a study including isolates with TR_34_, the proposed ECV was able to detect 92% (35/38) of mutant isolates [29]. Overall, Etest ECVs detected 61.2% (90/147) of all mutant isolates as NWT. More promising results were found with the Etest ECV of posaconazole (0.25 mg/L), which detected the majority of isolates (86%, 31/36) with different *cyp*51*A* substitutions including G54E/R/V/W, M220I/R/T/V/K, G448S and TR_34_/L98H [30,99]. Among the isolates that were not recognized were five mutants with M220I/R/T/V/K and G448S and other non-specified *cyp*51*A* substitutions with MICs ranging 0.023–0.25 mg/L. Overall, the proposed ECVs for posaconazole and itraconazole were able to detect 90.3% and 86.1% of mutant isolates with distinct mutations respectively, while results were less promising for the detection of mutant isolates for voriconazole (61.2%) (Table 3).

2.Echinocandins

ECVs for Etest and echinocandins, and more precisely for caspofungin and micafungin, have been determined in 2019 from three studies, including multi-laboratories with sufficient numbers of isolates tested [4,26,31]. Caspofungin ECVs have been defined for *A. fumigatus* (0.25 mg/L)*, A. flavus* (0.5 mg/L)*, A. terreus* (2 mg/L) and *A. niger* (0.25 mg/L)*,* [31], while a micafungin ECV has been defined only for *A. fumigatus* (0.016 mg/L) [4]. There are scarce data concerning the detection of mutants using Etest ECVs as very few isolates harboring resistance mechanisms have been described in the literature and susceptibility testing is not considered as an everyday practice in routine laboratory [102]. There is only one study with micafungin and caspofungin MIC data of laboratory mutants with known *fks* alterations with MTS [101]. An ECV of caspofungin was able to detect 3/3 of *fks* mutants, while micafungin’s ECV was able to predict 2/3 fks mutants as the third isolate had an MEC of 0.004 mg/L (Table 3).

3.Polyenes

Apart from CLSI and EUCAST ECV for amphotericin B, there is also a method-dependent ECV concerning Etest and *A. fumigatus* (2 mg/L), *A. flavus* (8 mg/L), *A. niger* (2 mg/L) and *A. terreus* (16 mg/L) [25]. Method-dependent ECVs for Etest are only available for these four *Aspergillus* species, but not for other filamentous fungi. Etest amphotericin B ECVs were consistently higher (one or two dilutions) or the same with CLSI method. *A. terreus* and *A. flavus* have high ECVs indicating intrinsic resistance for these species [25] in agreement with clinical cases with poor outcomes after amphotericin B therapy [93,103,104]. Since there is not any known resistance mechanism for *Aspergillus* and amphotericin B, studies including NWT isolates are not available in the literature.

### 3.1. Sensititre YeastOne

1.Triazoles

SYO ECVs of triazoles were determined in two large multicentre studies [29,30]. There are only ECVs for voriconazole (1 mg/L) and *A. fumigatus, A. flavus* and *A. terreus* and for posaconazole (0.06 mg/L) and *A. fumigatus*. It should be noted that the ECV for posaconazole and *A. fumigatus* is based on the unknown mutant status of the isolates. Applying the method-specific SYO ECV of voriconazole, 21/39 of mutant isolates with known *cyp*51*A* substitutions (TR_34_, G54E/R/W, M220I/K//R/T/V, G138C and I301T) were detected in one study [24] and 9/10 mutant isolates with TR_34_/L98H and TR_46_/Y121F T289A in another study [37]. In a third study, the voriconazole ECV of 1 mg/L was unable to detect two mutant isolates with G54R alteration and MIC 0.125 mg/L, whereas a single isolate with MIC 8 mg/L carrying the TR_34_/L98H *cyp*51*A* mutation has been recognized [93]. Overall, 67/116 (57.7%) of all mutant isolates have been characterized as NWT, indicating a concern about using SYO and voriconazole in order to detect mutant strains. The posaconazole SYO ECV was able to detect all 54/54 mutant isolates with the following *cyp*51*A* substitutions: TR_34_/L98H, TR_46_/Y121F T289A and G54R [37,105,106]. Considering the WT-UL (0.5 mg/L) that has been used in order to assess findings for itraconazole and *A. fumigatus*, 35/37 (94.5%) isolates with TR_34_/L98H, TR_46_/Y121F T289A and G54R mutations have been detected as NWT [37] in the majority of the studies, while there was a single study that included isolates with non-specified *cyp*51*A* substitutions, in which only 9/21 (42.8%) have been characterized as NWT [37] (Table 4). In this study, isolates with itraconazole and/or voriconazole CLSI MIC > 1 mg/L were submitted to *cyp*51*A* sequence analysis for the detection of azole-resistance-associated mutations and it was found that the isolates harbored TR_34_/L98H and TR_46_/Y121F T289A mutations. Overall, 45/65 (69.2%) of all isolates with various *cyp51A* mutations have been recognized. These results seem promising for posaconazole but not for voriconazole.

2.Echinocandins

ECVs for SYO and echinocandins have not been established till now, maybe due to the difficulty of the method used for the correct estimation of MEC. Recently, it has been proposed that the optimal conditions for SYO susceptibility testing of echinocandins is the use of an inoculum of 10^4^ CFU/mL, incubation for 20 h for *A. flavus* and 30 h for *A. fumigatus* and *A. terreus* and reading the first purple well. Agreement with CLSI reference method was good for micafungin (77–100%), with median (range) two-fold difference 0 (−1 to 2), −1 (−4 to 1) and −2 (−3 to −2), but poor for caspofungin (0–54%), with median (range) two-fold difference 3 (1 to 5), 3 (−4 to 4) and 2 (1 to 4) for *A. fumigatus*, *A. flavus* and *A. terreus*, respectively, indicating that SYO does not produce similar results as the reference method and therefore reference ECV should not be used for commercial test, while results were inconclusive for anidulafungin due to off-scale color endpoints [36]. To the best of our knowledge, there is no study where NWTs with known resistance mechanisms have been studied with SYO to echinocandins.

3.Polyenes

Regarding polyenes and especially amphotericin B, there is not a specific ECV for SYO for any filamentous fungi. In addition, there are no studies where SYO has been used for antifungal susceptibility testing of mutant isolates. Summarizing available studies, an MIC_90_ for 30 WT *A. fumigatus* isolates was 2 mg/L, while the same MIC_90_ was found for 24 *A. fumigatus* isolates harboring TR_34_ mutation as expected since *cyp51A* mutations do not affect amphotericin B susceptibility and for 10 *A. niger* isolates, whereas the MIC_90_ for 23 *A. flavus* strains was one dilution higher (4 mg/L) [105]. The MIC_90_ for 13 *A. terreus* which considered resistant to amphotericin was 2 mg/L indicating that SYO cannot differentiated amphotericin B susceptible from resistant species. In another study, an MIC = 1 mg/L for 2 *A. fumigatus* isolates harboring G54R mutations was reported, while an MIC = 2 mg/L for a single isolate with TR_34_/L98H mutation was also reported [106].

### 3.2. VIPcheck

VIPcheck is intended for the phenotypic detection of *A. fumigatus* resistance to itraconazole, voriconazole and posaconazole, in routine laboratories where the application of reference method is not possible [108,109]. Introduction of a 4-well plate into routine clinical laboratories has a huge impact on the early detection of azole resistance and subsequently benefits from more appropriate therapy for the patient [71]. It has been used for international *A. fumigatus* resistance prevalence study (SCARE), as well as other surveillance studies with overall good performance [50,51,107,110,111,112,113]. Categorical agreement with the reference method was 78.8–80% for voriconazole, 69.2–97.8% for itraconazole and 55.8–83.3% for posaconazole [50,107]. Lower categorical agreement was found for *A. fumigatus* cryptic species (Table 2). Moreover, the ability of VIPcheck to recognize isolates harboring TR mutations was excellent for itraconazole and voriconazole with 90.3% (159/176) and 80.1% (141/176) of mutants recognized as NWT, respectively. Results were less encouraging for posaconazole, with VIPcheck able to recognize 66% (115/176) of NWT isolates (Table 5).

### 3.3. Micronaut-AM

Micronaut’s ability to detect mutants has been assessed in two studies, with one showing ability to detect 89% (8/9) of CLSI azole NWT *Aspergillus* isolates [48] and the other one showing ability to detect 88% (14/16) of EUCAST azole NWT isolates with 15 harboring *cyp51A* mutations (eight TR_34_/L98H, 1 TR_46_/Y121F/T289A, three G54E/W and three other) [49]. In the last study, 1/15 (TR_46_/Y121F/T289A) was resistant to voriconazole (MIC = 4 mg/L), while 10/15 and 2/15 were resistant to itraconazole (MIC = 4 mg/L) and posaconazole (MIC = 0.5–1 mg/L), respectively [49]. In contrast, according to the authors, four *A. fumigatus* showed major and very major discrepancies. Three isolates were classified as WT by Micronaut for voriconazole (*n* = 2) and for itraconazole (*n* = 1) contrary to CLSI (very major), while one isolate was classified as NWT with Micronaut, whereas CLSI categorized it as WT (major) [48].

## 4. Conclusions

In conclusion, commercial methods can easily be applied in routine laboratories which do not have access to reference methods. However, they should be carefully applied following exactly the instructions of the manufacturers, taking into account the peculiarity of MIC reading for each drug and species for each method. VIPcheck method can be used to screen *Aspergillus* spp. for azole resistance and gradient concentration tests can be used to test susceptibility to each drug separately, whereas Sensititre YeastOne and Micronaut-AM can be used to test all drugs simultaneously. Among the few commercially available methods for antifungal susceptibility testing of molds, there are ECVs only for Etest and SYO and for the most common *Aspergillus* spp. (*A. fumigatus*, *A. flavus*, *A. terreus*, *A. niger* and *A. nidulans*). In addition, the ability of commercial tests to detect resistance is limited to *A. fumigatus* and triazole where isolates with known *cyp*51*A* substitutions have been studied. Thus, the performance of commercial tests in detecting resistance for other drugs and species including *A. fumigatus* cryptic species is unknown. Although in some cases the number of mutant isolates used to evaluate proposed ECVs is low, some conclusions can be made: (i) for Etest method, proposed ECVs for posaconazole, itraconazole and micafungin were able to detect >86% mutants of *A. fumigatus*, while ECV for voriconazole and caspofungin was less able to detect NWT with known *cyp*51*A* substitutions and *fks* alterations (<67%); (ii) for the SYO method, ECV for posaconazole exhibit encouraging results, recognizing all mutants, whereas ECV for voriconazole detect only 57.7% (67/116) of *A. fumigatus* isolates with *cyp*51*A* substitutions (Table 6). This is supported by results obtained from previous reports that showed isolates with TR_34_/L98H were characterized by a high itraconazole MIC (>8 μg/mL), variable susceptibility to voriconazole and cross-resistance to posaconazole [105]; (iii) finally, concerning VIPcheck as a method for early and reliable detection of azole resistance in routine clinical laboratory, where the usage of reference methodologies is not available, results were encouraging, with high categorical agreement especially for *A. fumigatus* sensu stricto isolates [50,107]. As not all mutations result in elevated MICs and NWT isolates may harbor unknown resistance mechanisms, one should be careful when evaluating performance of an MIC test based on their ability to detect mutants. However, a well described mutation that confer resistance to a specific drug should be captured by a certain test and failure to so means poor performance. Because most commercial methods have been developed for yeasts and then applied to molds, optimal performance cannot be guaranteed. There is a need for developing commercial methods for antifungal susceptibility testing of molds, taking into account the physiological (growth rate, metabolic activity, inoculum) and pharmacological (inhibition mode, killing activity) characteristics of each drug and species. Although the development of a commercial method that would produce the same MICs as the reference method for different drugs and species may be challenging, as long as the MICs of two methods are highly correlated, method-specific ECV could be used in order to improve categorical agreement. Finally, as each species have marked physiological characteristics, optimal conditions may be different among species even of the same genus. Further efforts are needed to develop an easy and fast commercial method for detecting resistance to many drugs among molds, particularly for species other than *A. fumigatus*.

## Figures and Tables

**Figure 1 jof-10-00214-f001:**
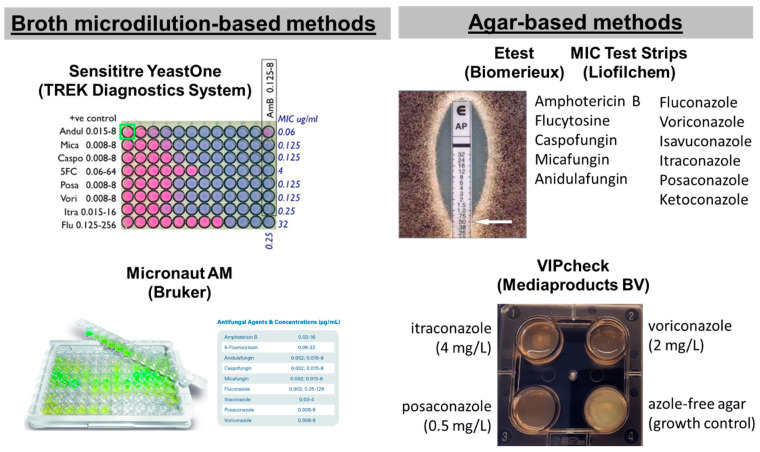
Commercial tests for antifungal susceptibility testing of filamentous fungi.

**Table 1 jof-10-00214-t001:** Method-dependent ECVs for clinically relevant filamentous fungi and available CLSI and EUCAST ECVs.

Drug and Species	Agent/Method-Dependent ECVs (mg/L)			
	CLSI ^a^	EUCAST ^b^	Etest ^c^	SYO ^d^
**Micafungin**				
*Aspergillus fumigatus* SC			0.016	
**Caspofungin**				
*Aspergillus fumigatus* SC	0.5		0.25	
*Aspergillus terreus* SC	0.125		2	
*Aspergillus flavus* SC	0.5		0.5	
*Aspergillus niger* SC	0.25		0.25	
**Isavuconazole**				
*Aspergillus fumigatus* SC	1	2		
*Aspergillus terreus* SC	1	(1)		
*Aspergillus flavus* SC	1	2		
*Aspergillus niger* SC	4	4		
*Aspergillus versicolor* SC	1			
*Aspergillus nidulans* SC		0.25		
**Voriconazole**				
*Aspergillus fumigatus* SC	1	1	0.5	1
*Aspergillus terreus* SC	2	2		1
*Aspergillus flavus* SC	2	2	0.5	1
*Aspergillus niger* SC	2	2	1	
*Aspergillus nidulans* SC		1		
*Fusarium solani* SC	32			
*Fusarium verticillioides* SC	4			
*Fusarium oxyporum* SC	16			
**Posaconazole**				
*Aspergillus fumigatus* SC	0.25	0.25	0.25	0.06
*Aspergillus terreus* SC	1	0.25	0.25	
*Aspergillus flavus* SC	0.5	0.5	0.5	
*Aspergillus niger* SC	2	0.5	0.5	
*Aspergillus nidulans* SC		0.5		
*Fusarium solani* SC	32			
*Fusarium verticillioides* SC	2			
*Fusarium oxyporum* SC	8			
*Lichtheimia corymbifera*	2			
*Mucor circinelloides*	4			
*Rhizopus arrhizus*	2			
*Rhizopus microsporus*	2			
**Itraconazole**				
*Aspergillus fumigatus* SC	1	1	2	
*Aspergillus terreus* SC	2	(0.5)		
*Aspergillus flavus* SC	1	1	1	
*Aspergillus niger* SC	4	2	4	1
*Aspergillus nidulans* SC		(1)	1	
*Fusarium solani* SC	32			
*Fusarium oxyporum* SC	32			
*Rhizopus arrhizus*	2			
**Amphotericin B**				
*Aspergillus fumigatus* SC	2	11	2 ^a^	
*Aspergillus terreus* SC	4	8	16 ^a^	
*Aspergillus flavus* SC	4	4	8 ^a^	
*Aspergillus niger* SC	2	(0.5)	2 ^a^	
*Aspergillus versicolor* SC	2			
*Aspergillus nidulans* SC		(4)		
*Fusarium (Gibberella) fujikuroi* SC		(8)		
*Fusarium solani* SC	8	(8)		
*Fusarium verticillioides* SC	4			
*Fusarium oxyporum* SC	8			
*Lichtheimia corymbifera*	2			
*Mucor circinelloides*	2			
*Rhizopus arrhizus*	4			
*Rhizopus microsporus*	2			

Values in brackets are tentative ECVs. Empty cells indicate that ECV have not been determined, SC: Species complex. ^a^ Data were retrieved from [5,14,15,24,25,26,27]. ^b^ Data were retrieved from [14,15,28]. ^c^ Data were retrieved from [4,25,26,29,30,31]. ^d^ Data were retrieved from [29].

**Table 2 jof-10-00214-t002:** Essential agreement ±2 two-fold dilutions (two-fold differences of median MIC) of the SYO, Etest, Micronaut-AM and VIPcheck tests compared to the CLSI method.

Method	Species	No. of Isolates	Anidulafungin	Micafungin	Caspofungin	Isavuconazole	Voriconazole	Posaconazole	Itraconazole	Amphotericin B	Ref.
**SYO**	** *A. fumigatus* ** **SC**	17		100% (0)	41% (3)						[36]
		21					100% (+1)	100% (0)	95.2% (−2)		[37]
		24							95.8% (NA)	95.8% (NA)	[38]
	***A. flavus* SC**	9		89% (−1)	0% (3)						[36]
		19					100% (+1)	100% (0)	94.7% (−1)		[37]
		23							78.3% (NA)	91.3% (NA)	[38]
	***A. terreus* SC**	13		77% (−2)	54% (2)						[36]
		12					100% (+1)	100% (−2)	91.7% (−1)		[37]
	***A. niger* SC**	7					100% (0)	85.7% (0)	100% (−1)		[37]
		7							100% (NA)	100% (NA)	[38]
	***A. nidulans* SC**	5					100% (+1)	100% (+1)	100% (−2)		[37]
	***Aspergillus* spp.**	61							90.2% (NA)	93.4% (NA)	[38]
	***Zygomycetes* ^a^**	45			93.3% (NA)		74.4% (NA)	81.8% (NA)	79.5% (NA)	29.5% (NA)	[39]
**Etest**	***A. fumigatus* SC**	74		100% (0)	96% (0)		95% (−1)	89% (−1)		99% (+1)	[40]
		24							45.8% (NA)	79.2% (NA)	[38]
		25			100% (NA)		100% (NA)	100% (NA)		100% (NA)	[41]
		26					65% (+2)				[42]
		38						82% (NA)			[43]
		21	100% (NA)	100% (NA)	90.4% (NA)						[44]
	Wild type	40				98 ^b^					[45]
	G54 alterations	10				100% ^b^					[45]
	M220 alterations	10				90% ^b^					[45]
	TR_34_/L98H	9				33% ^b^					[45]
	TR_46_/Y121F/T289A	10				100% (NA)^b^					[45]
	***A. flavus* SC**	29		100% (0)	100% (0)		100% (0)	83% (−1)		97% (+1)	[40]
		23							26.1% (NA)	60.9% (NA)	[38]
		21					100% (+1)				[42]
		18	100% (NA)	100% (NA)	77.8% (NA)						[44]
	***A. terreus* SC**	25		100% (0)	96% (0)		100% (−1)	64% (−2)		16% (+2)	[40]
		10					100% (+1)				[42]
		17	100% (NA)	100% (NA)	82.4% (NA)						[44]
	***A. niger* SC**	12		100% (0)	100% (0)		100% (−1)	83% (−1)		100% (+1)	[40]
		7							71.4% (NA)	100% (NA)	[38]
		13			100% (NA)		92% (NA)	100% (NA)		100% (NA)	[41]
		9					88% (+1)				[42]
		7	100% (NA)	100% (NA)	100% (NA)						[44]
	***A. nidulans* SC**	5					100% (0)				[42]
	***A. glaucus* SC**	4					50% (NA)				[42]
		4	100% (NA)	100% (NA)	50% (NA)						[44]
	***A. flavipes* SC**	2					100% (NA)				[42]
	***Aspergillus* spp.**	32					81% (+1)		72% (0)	69% (+5)	[46]
		61							42.6% (NA)	73.8% (NA)	[38]
		77					84.4% (+2)				[42]
		17						88% (NA)			[43]
	***Fusarium* spp.**	20					95% (0)	85% (−1)		90% (−1)	[47]
		34		100% (0)	100% (0)		100% (0)	100% (0)		94% (0)	[40]
	***Zygomycetes* ^a^**	45			100% (NA)		100% (NA)	81.8% (NA)	65.9% (NA)	54.5% (NA)	[39]
	***S. apiospermum* SC**	20		37% (−2)	63% (+1)		90% (−2)	90% (0)		80% (+1)	[40]
	***S. prolificans* SC**	5		100% (0)	100% (+1)		100% (0)	100% (0)		100% (0)	[40]
	***Scopulariopsis* spp.**	8		29% (4)	43% (+5)		100% (0)	100% (0)		75% (+1)	[40]
	***P. lilacium* SC**	20		100% (0)	100% (−1)		100% (0)	97% (−1)		100% (0)	[40]
**Micronaut-AM**	***Aspergillus* spp.**	78	99% (0)				90% (−1)		87% (−2)	100% (0)	[48]
	***A. fumigatus* SC ^c^**	77					58% (−2)	30% (−3)	62% (−2)	100% (0)	[49]
**VIPcheck ^d^**	***A. fumigatus* SC**	30					80%	83.3%	80%		[50]
	** *A. fumigatus* **	91					96.7%	73.6%	97.8%		[51]
	***A. fumigatus* cryptic species**	30					78.8%	55.8%	69.2%		[51]

^b^ MIC test Strip (MTS) MICs were compared with EUCAST MICs. MTS median MIC was one two-fold dilution lower than EUCAST median MIC for all isolates. ^a^ Included isolates of *Absidia* spp*., Cunninghamella* spp*., Mucor* spp*., Rhizomucor* spp*., Rhizopus* spp*., Syncephalastrum* spp. and as endpoint 24 h incubation time took into account. ^c^ Comparison between Micronaut-AM and EUCAST MICs. Numbers in brackets represent difference between modal MICs. ^d^ Categorical agreement is stated for VIPcheck^TM^. NA; not available. Empty cells indicate that there are no data.

**Table 3 jof-10-00214-t003:** Detection of *Aspergillus fumigatus* isolates harboring resistance mechanisms to triazoles/echinocandins with Etest dependent ECV.

Drug (Method-Specific ECV)	Mutations	MIC of All Mutant Isolates (mg/L)	No. Mutant Isolates with MIC > ECV	Refs.
**Isavuconazole (2 mg/L ^a^)**				
	G54 alterations G54E, G54R, G54V, G54W, G54R N248K	0.125–1	0/10	[45]
	M220 alterations M220K, M220T, M220V, M220I, M220I V101F	0.5–>32	2/10	[45]
	TR_34_/L98H	>32	9/9	[45]
	TR_46_/Y121F T289A	>32	10/10	[45]
	**Total**	**0.125–>32**	**21/39 (53.8%)**	
**Itraconazole (2 mg/L)**				
	*cyp51A* mutants	0.06–>16	78/81	[29]
	G54E, M220R, M220I, TR/L98H	12–>32	6/6	[99]
	*cyp51A* mutants	1.5	0/3	[100]
	G448S	0.5	0/1	[29]
	M220K	2	0/1	[29]
	I301T	≤0.06	0/1	[29]
	**Total**	**≤0.06–>32**	**8** **4/93 (90.3%)**	
**Voriconazole (0.5 mg/L)**				
	*cyp51A* mutants	0.06–>16	49/75	[29]
	G54E, M220R, M220I, TR/L98H	0.047–1.5	3/6	[99]
	*cyp51A* mutants	0.047	0/3	[100]
	I301T	<0.06	0/1	[29]
	TR_34_	0.125–>4	35/38	[29]
	G54E/R/W	≤0.06–0.5	0/12	[29]
	M220I/K//R/T/V	0.125–>4	3/11	[29]
	G138C	0.25	0/1	[29]
	**Total**	**0.047–>16**	**90/147 (61.2%)**	
**Posaconazole (0.25 mg/L)**				
	G54E, M220R, M220I, TR/L98H	1–>16	6/6	[99]
	TR_34_/L98H	0.5–>16	13/13	[30]
	G54E/R/V/W	2–>16	5/5	[30]
	M220I/R/T/V/K	0.25–>16	3/4	[30]
	G448S	0.25–1	4/5	[30]
	*cyp51A* mutants	0.023	0/3	[100]
	**Total**	**0.023–>16**	**31/36 (86.1%)**	
**Micafungin (0.016 mg/L) ^b^**				
	*fks* alterations S678P	0.004–1	2/3	[101]
	**Total**	**0.004–1**	**2/3** **(66.7%)**	
**Caspofungin (0.25 mg/L) ^b^**				
	*fks* alterations S678P	2–8	3/3	[101]
	**Total**	**2–8**	**3/3** **(100%)**	

^a^ As no method-specific ECVs have been determined for isavuconazole, we used an MIC two two-fold dilutions higher than the median MIC of WT isolates [45]. ^b^ MIC tests trips by Lioflilchem were used.

**Table 4 jof-10-00214-t004:** Detection of *Aspergillus fumigatus* isolates harboring resistance mechanisms to triazoles/echinocandins with Sensititre YeastOne-dependent ECV.

Drug (Method-Specific ECV)	Mutations	MIC of All Mutant Isolates (mg/L)	No. Mutant Isolates with MIC > ECV	Refs.
**Itraconazole (0.5 mg/L ^a^)**				
	TR_34_/L98H	2–>16	5/5	[37]
	TR_46_/Y121F T289A	0.5–1	4/5	[37]
	TR_34_	>16	24/24	[105]
	G54R	1	2/2	[106]
	TR_34_/L98H	>16	1/1	[106]
	*cyp51A* mutants	0.125–>8	9/21	[37]
	TR_34_/L98H, S297T, F495I, TR_46_/Y121F T289A	0.125–>16	4/7	[107]
	**Total**	**0.125–>16**	**45/65** **(69.2%)**	
**Voriconazole (1 mg/L)**				
	*cyp51A* mutants	0.125–>16	21/39	[29]
	I301T	0.125	0/1	[29]
	TR_34_	1	0/1	[29]
	G54E/R/W	0.125–0.5	0/5	[29]
	M220I/K//R/T/V	0.25–0.5	0/5	[29]
	G138C	0.25	0/1	[29]
	TR_34_/L98H	1–8	4/5	[37]
	TR_46_/Y121F T289A	8–>8	5/5	[37]
	TR_34_	1–8	>12/24 *	[105]
	G54R	0.125	0/2	[106]
	TR_34_/L98H	8	1/1	[106]
	*cyp51A* mutants	0.125–>8	10/21	[37]
	TR_34_/L98H, S297T, F495I, TR_46_/Y121F T289A	1–>8	5/7	[107]
	**Total**	**0.125–>16**	**67/116** **(57.7%)**	
**Posaconazole (0.06 mg/L)**				
	TR_34_/L98H	0.5–1	5/5	[37]
	TR_46_/Y121F T289A	1	5/5	[37]
	TR_34_	0.25–1	24/24	[105]
	G54R	1	2/2	[106]
	TR_34_/L98H	1	1/1	[106]
	*cyp51A* mutants	0.5–1	10/10	[37]
	TR_34_/L98H, S297T, F495I, TR_46_/Y121F T289A	0.125–0.25	7/7	[107]
	**Total**	**0.25–1**	**54/54** **(100%)**	

* We assumed >12/24 because MIC_50_ is 2 mg/L. ^a^ As no method-specific ECVs have been determined for itraconazole, we used an MIC two two-fold dilutions higher than the median MIC of WT isolates [105].

**Table 5 jof-10-00214-t005:** Detection of *Aspergillus fumigatus* isolates harboring resistance mechanisms to triazoles/echinocandins with VIPcheck^TM^.

Drug	Mutations	MIC of All Mutant Isolates ^a^ (mg/L)	No. Mutant Isolates Detected	Refs.
**Itraconazole**				
	TR_34_/L98H (28/30)	2–>16	25/30	[50]
	TR/L98H	>8	3/4	[111]
	TR_34_/L98H G54R, N248K	1–>16	3/3	[106]
	TR_34_/L98H, TR_46_/Y121F T289A	1–>8	47/47	[110]
	TR_34_/L98H, TR_46_/Y121F T289A, G54, M220	0.5–>16	32/39	[71]
	TR_34_/L98H, TR_46_/Y121F, TR_53_ T289A, G54, M220, P216, G138, G448	NA	29/33	[109]
	*cyp51A* mutations	>16	20/20	[109]
	**Total**	**0.5–>32**	**159/176** **(90.3%)**	
**Voriconazole**				
	TR_34_/L98H (28/30)	2–4	23/30	[50]
	TR/L98H	4	1/4	[111]
	TR_34_/L98H G54R, N248K	0.125–8	2/3	[106]
	TR_34_/L98H, TR_46_/Y121F T289A	2	47/47	[110]
	TR_34_/L98H, TR_46_/Y121F T289A, G54, M220	0.125–>16	23/39	[71]
	TR_34_/L98H, TR_46_/Y121F, TR_53_ T289A, G54, M220, P216, G138, G448	NA	29/33	[109]
	*cyp51A* mutations	0.5–>16	16/20	[109]
	**Total**	**0.03–>16**	**141/176** **(80.1%)**	
**Posaconazole**				
	TR_34_/L98H (28/30)	0.5–1	7/30	[50]
	TR/L98H	0.5	1/4	[111]
	TR_34_/L98H G54R, N248K	1	2/3	[106]
	TR_34_/L98H, TR_46_/Y121F T289A	0.25–1	47/47	[110]
	TR_34_/L98H, TR_46_/Y121F T289A, G54, M220	0.125–>16	15/39	[71]
	TR_34_/L98H, TR_46_/Y121F, TR_53_ T289A, G54, M220, P216, G138, G448	NA	29/33	[109]
	*cyp51A* mutations	0.5–2	14/20	[109]
	**Total**	**0.125–>16**	**115/174** **(66%)**	

^a^ MIC of the isolates is based according to reference methods.

**Table 6 jof-10-00214-t006:** Summary of the ability of each method to detect resistance of NWT *Aspergillus fumigatus* isolates harboring resistance mechanisms to triazoles/echinocandins. Percent of mutant isolates classified as NWT (N mutants/total isolates) are shown for each drug and method.

Method	Isavuconazole	Itraconazole	Voriconazole	Posaconazole	Micafungin	Caspofungin
**Etest**	53.8% (21/39)	90.3% (84/93)	61.2% (90/147)	86.1% (31/36)	66.7% (2/3)	100% (3/3)
**SYO**		69.2% (45/65)	57.7% (67/116)	100% (54/54)		
**VIPcheck**		90.3% (159/176)	80.1% (141/176)	66% (115/174)		
**Micronaut-AM**	88% (22/25)					

Empty cells indicate that there are no data.

## Data Availability

Not applicable.

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
