# Peer review of "Commercial Methods for Antifungal Susceptibility Testing of Saprophytic Molds: Can They Be Used to Detect Resistance?"

_jof, 2024, doi:10.3390/jof10030214_

Round 1

Reviewer 1 Report

Comments and Suggestions for Authors

Dear authors

The article you provided is globally well written and clear (which was a challenge given the complexity of the depiction of the various methods for antifungal susceptibility testing in Molds)

I have no major corrections to ask for

The only minor issue i see is the Table 2 which is awkward to read, (especially with a lot of misplaced brackets and lines cut) This table 2 should be revised and clarified

Regards

Author Response

The only minor issue i see is the Table 2 which is awkward to read, (especially with a lot of misplaced brackets and lines cut) This table 2 should be revised and clarified.

Response: Brackets and lines are now corrected

Reviewer 2 Report

The work of Paranos and cols. is an interesting and exhaustive review regarding the commercial methods for antifungal susceptibility testing of clinically relevant molds. In general, the work is very informative and brings the state of art of the aforementioned topic. Moreover, the manuscript is adequately structured and well written.

N/A.

Author Response

No comments

Reviewer 3 Report

Dear Authors:

I just have a few minor corrections and a suggestion.

Please see detailed comments

Best regards

P    1. Please correct spp. It should not be written in italics. (see the whole text)

2.       In point 3. (Commercial methods…) Please correct the following statement: “Therefore, commercially available and ready-to-use methods could be a better alternative for the routine clinical microbiology laboratory as far as they are able to produce similar results with reference standards [27]. voriconazole, itraconazole and posaconazole”

3.       In page 15 Echinocandins. Please correct 20 h and 30 h (the figure should be separated from the unit)

4.       It would be interesting to present in a table the capacity of each method to detect resistance to different antifungals and thus be able to compare more easily which method is more convenient in each case.

Author Response

  1. Please correct spp. It should not be written in italics. (see the whole text)

Response: Done

  1. In point 3. (Commercial methods…) Please correct the following statement: “Therefore, commercially available and ready-to-use methods could be a better alternative for the routine clinical microbiology laboratory as far as they are able to produce similar results with reference standards [27]. voriconazole, itraconazole and posaconazole”

Response: Done

  1. In page 15 Echinocandins. Please correct 20 h and 30 h (the figure should be separated from the unit)

Response: Done

  1. It would be interesting to present in a table the capacity of each method to detect resistance to different antifungals and thus be able to compare more easily which method is more convenient in each case.

Response: Table 6 has been added.

Reviewer 4 Report

I find the article very interesting because it describes the importance, as well as the disadvantages or limitations of each of the commercial antifungal susceptibility methods present on the market and which are mainly based on the two reference methods: CLSI and EUCAST. But the authors should take into account that the knowledge we have about these methods is that they are guidelines to be used routinely in Mycology laboratories. Therefore, I can say that this has already been studied and said, in short it is not something new. What I think is interesting to do an exhaustive review of the topic would be to take into account the genes and mutations that are involved in resistance, this leads me to conclude that the most important thing would be to direct the review to virulence factors responsible for the resistance that currently present fungi to antifungals.

I would like to say that there are many errors related to lack of meaning of abbreviations.

When they talk about molds in the title, they are referring to all those filamentous fungi, so we have to include dermatophytes, systemic mycotic fungi, etc.

We need to talk a little about the standardization of reference methods and not just mention them, this will help us understand a little more about commercial methods.

They do not talk about the limitations as such, they could mention as an example the feasibility of acquiring these tests or not.

Mention whether commercial methods are easy to use and whether they need to be calibrated.

etc

​

It is important to comment that they should improve the organization of the tables for their understanding.

The figures are very small, they should be bigger.

Author Response

I find the article very interesting because it describes the importance, as well as the disadvantages or limitations of each of the commercial antifungal susceptibility methods present on the market and which are mainly based on the two reference methods: CLSI and EUCAST. But the authors should take into account that the knowledge we have about these methods is that they are guidelines to be used routinely in Mycology laboratories. Therefore, I can say that this has already been studied and said, in short it is not something new. What I think is interesting to do an exhaustive review of the topic would be to take into account the genes and mutations that are involved in resistance, this leads me to conclude that the most important thing would be to direct the review to virulence factors responsible for the resistance that currently present fungi to antifungals.

Response: The purpose of the review was not to describe the virulence factors and all mutations which are connected with filamentous fungi, but to summarize known mutations associated with resistance.

I would like to say that there are many errors related to lack of meaning of abbreviations.

Response: We revised abbreviations both in abstract and the main text.

When they talk about molds in the title, they are referring to all those filamentous fungi, so we have to include dermatophytes, systemic mycotic fungi, etc.

Response: We have now stated “saprophytic molds”.

We need to talk a little about the standardization of reference methods and not just mention them, this will help us understand a little more about commercial methods.

Response: We have now added the information about the reference method.

They do not talk about the limitations as such, they could mention as an example the feasibility of acquiring these tests or not.

Response: Limitations are now mentioned in lines 119-122.

Mention whether commercial methods are easy to use and whether they need to be calibrated.

Response: This is now stated in lines 122-124.

Reviewer 5 Report

The manuscript “Commercial Methods for Antifungal Susceptibility Testing of Molds: Can They Be Used to Detect Resistance?” is a review of studies involving commercial methods to evaluate the in vitro susceptibility of filamentous fungi and their ability to detect non-wild-type (NWT) isolates, according to the available epidemiological cut-off values (ECVs). It is an interesting work, considering that the number of invasive fungal infections has been increasing in recent years; in addition to the appearance of cryptic species associated with resistance to antifungals, they constitute a serious public health problem worldwide. However, I have some comments, which I list below.

Comments

It seems to me that the order of the article is not appropriate, so I suggest that the mechanisms of resistance to antifungals (5. Echinocandins, 6. Triazoles, and 7. Polyenes) should be changed before section “4. ReviewPurpose.”

All text should be carefully reviewed to correct typographical errors.

Introduction

In section 2, indicate the meaning of “ECV” when it is mentioned for the first time in the text.

At the bottom of Table 1, indicate the meaning of “SC”

At the bottom of table 1: Change “fro.m” to “from”

In section 3, correct the end of this paragraph: “Therefore, commercially available and ready-to-use methods could be a better alternative for the routine clinical microbiology laboratory as far as they are able to produce similar results with reference standards [27]. voriconazole, itraconazole and posaconazole

I suggest decreasing the font size in Table 2 to ensure that the text in each row is on a single line. Additionally, correct that the data in each column corresponds to the column header since the data is disordered in the current format.

Discussion

I believe the authors should discuss the advantages and disadvantages of the methods included in this review to highlight the most useful.

 References

Authors must carefully review references and strictly follow the format requested by the journal.

Author Response

It seems to me that the order of the article is not appropriate, so I suggest that the mechanisms of resistance to antifungals (5. Echinocandins, 6. Triazoles, and 7. Polyenes) should be changed before section “4. ReviewPurpose.”

Response: The purpose of the review was also to describe the mechanism of actions and resistance and summarize known mutations in target and other genes that are associated with resistance. This is now staed in lines 166-168. Therefore, we did not change the sequence of the sections .

All text should be carefully reviewed to correct typographical errors.

Response: Done

Introduction

In section 2, indicate the meaning of “ECV” when it is mentioned for the first time in the text.

Response: Corrected in the abstract. In the main text we describe epidemiological cutoff values (ECOFFs for the EUCAST reference method and ECVs for the CLSI reference method) in line 75.

At the bottom of Table 1, indicate the meaning of “SC”

Response: Done

At the bottom of table 1: Change “fro.m” to “from”

Response: Done

In section 3, correct the end of this paragraph: “Therefore, commercially available and ready-to-use methods could be a better alternative for the routine clinical microbiology laboratory as far as they are able to produce similar results with reference standards [27]. voriconazoleitraconazole and posaconazole

Response: Done

I suggest decreasing the font size in Table 2 to ensure that the text in each row is on a single line. Additionally, correct that the data in each column corresponds to the column header since the data is disordered in the current format.

Response: Done

Discussion

I believe the authors should discuss the advantages and disadvantages of the methods included in this review to highlight the most useful.

Response: Done

References

Authors must carefully review references and strictly follow the format requested by the journal.

Response: Done

Round 2

Reviewer 4 Report

This is ok

This is ok